# Effect of Particle Size on Microstructure and Element Diffusion at the Interface of Tungsten Carbide/High Strength Steel Composites

**DOI:** 10.3390/ma12244164

**Published:** 2019-12-11

**Authors:** Hongmei Zhang, Hongnan Li, Ling Yan, Chao Wang, Fangfang Ai, Yan Li, Na Li, Zhengyi Jiang

**Affiliations:** 1School of Material and Metallurgy, University of Science and Technology Liaoning, Anshan 114000, China; lihongnan7270@163.com (H.L.); wangchao1908@163.com (C.W.); huatsing2006@163.com (N.L.); 2The State Key Laboratory of Metal Material for Marine Equipment and Application, Anshan Iron and Steel Research Institute of AnGang Group, Anshan 114009, China; yanling_1101@126.com (L.Y.); aifangfang@163.com (F.A.); 2323liyan@sina.com (Y.L.); 3Technology Research Institute of Benxi Steel Group, Benxi 117000, China; 4School of Mechanical, Materials, Mechatronic and Biomedical Engineering, University of Wollongong, Wollongong 2522, Australia

**Keywords:** composites, microstructure, interface, particle size

## Abstract

The microstructure and micro-hardness of tungsten carbide/high strength steel (WC/HSS) composites with different particle sizes were analyzed by optical microscopy (OM), scanning electron microscopy (SEM), ultra-high temperature laser confocal microscopy (UTLCM) and micro-hardness testing. The composites were prepared by cold pressing and vacuum sintering. The results show that WC density tends to increase as the average grain size of WC decreases and the micro-hardness of WC increases with the decrease of WC particle size. The micro-hardness of WC near the bonding interface is higher than that in other regions. When the particle size of WC powder particles is 200 nm, a transition layer with a certain width is formed at the interface between WC and HSS, and the combination between the two materials is metallurgical. The iron element in the HSS matrix diffuses into the WC structure in contact with it, resulting in a fusion layer of a certain width, and the composite interface is relatively well bonded. When the average particle size of WC powder is 200 nm, W, Fe and Co elements significantly diffuse in the transition zone at the interface. With the increase of WC particle size, the trend of element diffusion decreases.

## 1. Introduction

Tungsten carbide (WC) materials are widely used in micro-drills, nuclear refractory parts, medical equipment, cutting tools and wear-resistant parts due to their attractive properties such as high hardness, high melting point, high density, high wear resistance and corrosion resistance [1,2,3,4]. However, the main raw material for WC production, tungsten, is an extremely scarce and precious resource, which has important strategic significance in China [5]. Therefore, it is necessary to develop a composite material having excellent comprehensive performance based on this need for the economical and efficient use of resources.

WC carbide is an intrinsically brittle material. Components made of WC carbide break easily. Especially in precision micro-drill applications, slight deviations between the machine and the micro-drill can cause the micro-bit to break, resulting in loss of tools and workpieces [6,7]. Therefore, a material having high strength and high fracture toughness can be used as the supporting core of the WC material. To save rare resources and reduce production costs, the external use of a coating of ultra-fine grain tungsten carbide material can be used as the wear-resistant layer, so that it has superior hardness, wear resistance and fracture strength. The high-strength steel used as internal core material can simultaneously improve the strength and fracture toughness of the composite. The development of such tungsten carbide/high strength steel composites is of great significance. Composite materials based on tungsten carbide as reinforcing particles and steel materials have attracted growing attention [8]. The combination of tungsten carbide reinforced steel matrix composites is divided into integral composite and surface composite. The main preparation methods include powder metallurgy, the in-situ composite method (integral composite), laser cladding methods, surfacing method and the casting infiltration method (surface composite) [9]. In order to solve the problem of mismatch between thermal expansion coefficients between the tungsten carbide/steel composite matrix and the reinforcement, many scholars have carried out research to improve the interface of the composite. Chen [10] prepared WC/Cr15 steel-based surface composites using the vacuum casting process, and examined the effect of adding tungsten-iron powder on the interfacial microstructure and properties of composites. Sui [11] et al. fabricated WC particulate reinforced steel matrix surface composites using the casting infiltration method. The effect of Ni addition on the interfacial structure of cast infiltration layer was studied. Feng [12] obtained the WC iron-based composite material by sintering the pure iron powder and tungsten carbide mixed powder at 1500 °C for 6 min at atmospheric pressure. C.M. Fernandes [13] prepared WC-AISI304 stainless steel cemented carbide using the powder metallurgy method, and studied the mechanical properties, thermal properties and corrosion resistance of the material. Chun-Ming Lin [14] performed a tungsten carbide reinforced steel matrix composite by laser heating using raw austenite 304L stainless steel powder with a particle size of 100–150 μm and tungsten carbide powder with a particle size of 30–50 μm. Further, it was found that during composite material processing, the tungsten carbide powder is dissolved in the melt and then solidified to form metal carbides such as (Fe, W) 3C and M7C3 (M = Cr, Fe). The mechanical properties have been improved by the formation of these carbides, such as hardness and fracture toughness of stainless steel.

In recent years, research on WC particle reinforced steel matrix composites mainly focused on the influence of sintering temperature, holding time, preformed pressure and other process parameters, on the microstructure, properties and interface of the composite. These studies were carried out under conditions in which the particle size of the tungsten carbide particles was kept constant, and the influence of the particle size of the tungsten carbide particles on the microstructure, properties and interface of the composite material was not considered.

At present, few studies examine the influence of the particle size of tungsten carbide particles on the microstructure, properties and interface of tungsten carbide/steel composites. In addition, most of the existing research has focused on the manufacture of macro-scale parts [15], while tungsten carbide/steel composites for small parts have rarely been reported.

## 2. Experimental Materials and Methods

Tungsten carbide/high strength steel composites were prepared by cold pressing and vacuum sintering with self-made dies. The schematic diagram of the self-made die is shown in Figure 1a. The inner diameter of the die is 3 mm and the outer diameter is 6 mm. The raw material powders used in the experiment were: four kinds of WC powders having particle diameters of 100 nm, 200 nm, 300 nm, and 500 nm and Co powders with particle size of 25 nm. The high-strength steel core material is M2 high-speed steel with a diameter of 1 mm (microstructure as shown in Figure 1b). The specific experimental steps are as follows: Firstly, the high-speed steel core with diameter of 1 mm is polished, and then the polished steel core is fixed to the center of the die by the base. WC powder and Co powder with different particle sizes are mixed evenly in the mass ratio of 9:1 and filled into the die. Mixed powder and high-strength steel core are compacted together by MTS810 fatigue testing machine with a load of 1400 MPa. Then the whole set of dies is sintered in the vacuum tube furnace (GR. TF60). In order to avoid material oxidation in the sintering process, vacuum is pumped before heating, and then argon is injected to ensure exhaustion of air. The above steps were repeated three times. The whole sintering process is carried out under the protection of argon. The sintering temperature is set to 1300 °C, holding for 90 min, and then cooled to room temperature with the furnace after the heat preservation. Microstructure and morphology of the composites were observed by optical microscopy (OM VHX-500, Keyence (China) Co., Ltd., Shanghai, China) and scanning electron microscopy (SEM Zeiss-IGMA HD, Jena, Germany), and the interface of the composites was analyzed by surface scanning. The microstructures of the composites during heating process were observed and analyzed by VL2000DX ultra-high temperature laser confocal microscopy (Yonekura MFG Co., LTD, Osaka, Japan). Based on the microstructure of WC-10Co under different particle sizes and combined with Image J software, its density was analyzed [16]. The micro-hardness of the composites was measured on a digital micro-hardness tester (Q10M, Qness GmbH, Salzburg, Austria). The loading force was 0.1 kg and the loading time was 10 s. The method of dotting is selected from the high-strength steel center along the radius direction, and one point is taken out every 0.2 mm, and seven points are punched for each sample.

## 3. Results and Analysis

### 3.1. WC Microstructure Analysis

In order to study the effect of particle size of tungsten carbide on the microstructures of tungsten carbide/high strength steel composites, ultrafine tungsten carbide powders with average particle sizes of 100, 200, 300 and 500 nm were selected. WC-Co cemented carbide with nano-sized ultra-fine WC powder has good mechanical properties and cutting properties and is widely used in metal cutting and PCB processing industries [17]. Figure 2 shows a SEM image of the original WC powder morphology. As can be seen from Figure 2a–d, the WC powder has a spherical or polyhedral shape. Figure 2a–c have a relatively uniform particle size distribution of tungsten carbide powder. However, due to the small particle size and large specific surface area, a certain “agglomeration” occurs under the action of surface energy. “Agglomeration” refers to the phenomenon in which a plurality of particles adhere together to become agglomerated. The main reason for agglomeration is the interaction of surface energy such as charge, moisture and van der Waals force. The finer the particle, the greater the surface energy and the chance of assemblage [18]. When the average particle size of WC is 300 nm, the agglomeration phenomenon is the most serious, as shown in Figure 2c. When the average particle size of WC is 500 nm, the distribution of WC particle size varies greatly. The diameter of individual WC particles reaches more than 1μm, as shown in Figure 2d.

Figure 3 is a SEM image of the microstructure of WC-10Co cemented carbide sintered at 1300 °C with different particle sizes. The grain size of WC was measured by truncation method. When the average particle size of WC was 100 nm, the average grain size of sintered WC was 0.566 μm. The grains of WC are polygonal with obvious grain boundaries. The gray phase Co distributes uniformly around the grains of WC as shown in Figure 3a. When the average particle size of WC is 200 nm, the average grain size of sintered WC is 0.658 μm, and the WC grain is polygonal, but the grain boundary is not obvious, and the distribution of Co phase is not very uniform as shown in Figure 3b. When the WC average particle diameter is 300 nm or 500 nm, the surface of the WC structure is uneven, and a large number of pores (black portions in 3c) exist. This is because a higher sintering temperature is required when the WC particles have a larger particle size, and the temperature does not reach the sintering temperature, and the WC structure is not densified.

Figure 4 is a dependence curve showing the change in density of tungsten carbide cemented carbides of different particle sizes after sintering at the same sintering temperature. It can be seen that the density of the tungsten carbide cemented carbide tends to decrease as the particle size of the tungsten carbide increases. When the average particle size of the tungsten carbide powder is 100 nm, the density of the cemented carbide after sintering is reduced to 91.22%. When the average particle size of the tungsten carbide powder is 200 nm, the density of the cemented carbide after sintering is reduced to 79.58%. When the average particle diameter of the tungsten carbide powder is 300 nm or 500 nm, the cemented carbide after sintering has a low density of 60% or less. It is indicated that when the average particle size of the tungsten carbide powder is 300 nm or 500 nm, a higher sintering temperature is required to complete the densification process. It is generally believed that the particle size of the original WC particles affects the sintering temperature of the WC cemented carbide, such that the finer the particles, the lower the sintering temperature required and the lower the temperature at which densification is accomplished. Therefore, the finer the average particle size of the original WC powder, the higher the degree of densification after sintering at the same temperature. At the same time, the center distance of the two tungsten carbide particles is shortened as the particles are reduced, so that the fine-grained powders are more closer to each other when sintered. The finer the powder particle size, the larger the specific surface area, and the higher the dissolution rate after the appearance of solid phase diffusion and liquid phase.

### 3.2. Effect of Particle Size of Tungsten Carbide Particles on Micro-Hardness of Tungsten Carbide/High Strength Steel Composites

Figure 5 and Figure 6 show the Vickers hardness distribution curves of tungsten carbide/high strength steel composites sintered at 1300 °C and 1320 °C after WC mixing, charging and pressing at different particle sizes. The distribution diagram of the measured points is shown in Figure 5a. As can be seen from Figure 5 and Figure 6, the hardness of the sintered WC cemented carbide decreases with increasing WC particle size under certain conditions of sintering temperature. This is because the hardness of WC cemented carbide is mainly related to the density and the grain size of WC: the higher the density, the greater the hardness, the smaller the grain size, the greater the hardness. In this experiment, the finer the particle size of the original WC powder, the higher the WC density, and the larger the grain size of the WC microstructure after sintering. It can be seen from Figure 5 that the hardness of the tungsten carbide side is significantly higher than that of the high-strength steel matrix, and the hardness value changes significantly at the interface between the tungsten carbide and the high-strength steel matrix, and there is a “jump” phenomenon. When the particle size of tungsten carbide is 100 nm, the hardness of WC after sintering at 1300 °C reaches 1300HV0.1-1400HV0.1. When the particle size of tungsten carbide is 200 nm, the hardness value of WC reaches 1299 HV0.1.

It can be seen from Figure 6 that when the particle size of tungsten carbide is 100 nm, the maximum hardness of WC is about 1680 HV0.1 after sintering at 1320 °C. When the tungsten carbide particle size is 200 nm, the WC hardness value is 1400 HV0.1 after sintering at 1320 °C. Srivatsan et al. [19] prepared tungsten carbide cemented carbide using a spark plasma sintering process with 0.2 μm, 0.8 μm and 1.2 μm tungsten carbide raw powders. The effect of particle size of tungsten carbide on the microstructure and micro-hardness of cemented carbide was studied. Their results show that the increase of powder particle size will lead to the decrease of the hardness of tungsten carbide cemented carbide. When the particle size of tungsten carbide powder is 0.2 μm, the average hardness value is 1377HV0.5, which is not greatly different from the results of this study. The high-speed steel material in the core has little change with the WC particle size, which is about 500 HV0.1. The WC hardness near the interface is higher than other positions, which is beneficial as it strengthens the interface bonding strength.

### 3.3. Composite Interface Microstructure and Element Diffusion

Figure 7 is a scanning electron microscopy image of a tungsten carbide/high strength steel composite prepared by different particle sizes of tungsten carbide particles at 1300 °C. The average particle sizes of the original tungsten carbide powder are shown in Figure 7a–d are 100, 200, 300 and 500 nm, respectively. It can be seen from the graph that when the average particle size of WC is 100 nm, the interface defect of WC/HSS composite is obvious, and there are obvious wide cracks at the interface, as shown in Figure 7a. This is because the finer the WC particles, the worse the fluidity, the worse the diffusion of elements between high temperature steel and high strength steel, and the worse the diffusion of iron elements to WC. As the thermal expansion coefficients between the two materials are quite different, the stress concentration phenomenon will occur at the interface during cooling to room temperature, leading to the formation of more obvious cracks. When the average particle size of WC is 200 nm, the iron element in the matrix of high strength steel diffuses into the contact tungsten carbide structure and produces a certain width of fusion layer, as shown in Figure 7b. This is because at this temperature, the high strength steel has begun to melt slowly, so the diffusion of Fe element into tungsten carbide structure reacts with tungsten carbide. When the average particle size of tungsten carbide powder is 300 nm or 500 nm, there is no transition layer at the interface of composites, as shown in Figure 7c,d. This is due to the fact that WC has not been densified at this sintering temperature, and the particle size of WC reinforcement is relatively large. The bonding of WC reinforcement mainly depends on the friction force produced by the shrinkage of rough reinforcement surface and matrix. The bonding of the two materials is close to mechanical bonding.

In addition, when the particle size of tungsten carbide powder is 500 nm, a large number of circular “black spots” can be observed in the high strength steel near the interface as shown in Figure 7d. The results of Energy-Dispersive X-ray Spectroscopy (EDS) analysis are shown in Figure 8. It can be found that the black spots are mainly composed of impurities of C, O, Na, Si, K and S elements. This may be due to the inclusion of other impurities in the sample preparation process.

Figure 9 is the SEM micrograph and EDS image of the interface of WC/HSS composites prepared using tungsten carbide powder with particle sizes of 200 nm and 500 nm at 1300 °C. Using the same sintering temperature of 1300 °C, the left side of the image is WC and the right side is M2 high-speed steel. The main elements detected by surface scanning are Fe, W, Cr and Co. When the average particle size of WC powder is 200 nm, it can be seen that W, Fe and Co elements have obvious diffusion at the interface in the transition region, as shown in Figure 9a. Shan [20] studied the microstructure of the composite interface of tungsten carbide-reinforced surface composites with different particle sizes. It was found that the solubility of tungsten carbide particles in the matrix decreased with the increasing of the size of tungsten carbide particles.

In order to observe the change trend of the main elements of tungsten carbide/high-strength steel composites at the interface, the composite sample with sintering temperature of 1300 °C was selected for line scan analysis by the energy spectrometer (EDS) attached to the scanning electron microscope. Figure 10 is a line scan analysis map of Fe, W and Co elements. It can be seen from the image that the scanning area can be divided into three parts from left to right, namely a high-strength steel base, a transition layer and a tungsten carbide reinforcement, and the transition layer has a width of about 25–30 μm. The Fe element is mainly present in the high-strength steel matrix. There are two changes from the left to the right. First, the Fe content from the high-strength steel matrix to the interface is rapidly reduced, and there is a certain amount of Fe in the transition zone. Then, from the transition zone to the tungsten carbide reinforcement, the content of Fe is reduced again. Therefore, in this process, the Fe diffuses into the tungsten carbide structure. On the contrary, W changes from the right side to the left side only once. From the tungsten carbide structure to the transition layer, content of W gradually decreases, and the content of W in the high-strength steel matrix area is extremely small, almost negligible. The content of Co is higher in the transition zone and diffuses to the high-strength steel matrix. The Fe, W and Co coexist in the transition layer region, and the three elements react in this region, resulting in new phases and good metallurgical bonding of the composites.

Figure 11 shows the line scanning pattern at the interface of the composite with tungsten carbide particle size of 500 nm. The three elements tested are Fe, W and Co. It can be seen from the Figure 11 that the interface between the matrix of composite high-strength steel and tungsten carbide reinforcement is obvious, and there is no transition layer. At the same time, Fe decreases sharply at the interface, and there is a small amount of Fe in the right tungsten carbide region, and the content change tends to be gentle. W mainly exists in the right tungsten carbide structure. Its content decreases sharply at the right-to-left interface and is negligible in the right high-strength steel matrix. Co element shows a downward trend from right to left, and the overall change is gentle, which proves that Co element diffuses into high strength steel matrix. Therefore, when the particle size of tungsten carbide is 500 nm, Fe and Co elements diffuse slightly. However, compared with Figure 9 (tungsten carbide particle size 200 nm), there is no transition layer at the interface of the composites, which proves that there is no interfacial reaction and new phase in the composites.

### 3.4. Analysis of Experimental Results of High Temperature Laser Confocal Microscopy (CLSM)

The traditional methods used to study the microstructure of metal materials mainly includes optical microscopy, scanning electron microscopy and transmission electron microscopy. However, these methods can only observe the microstructure of the material in static state and cannot observe the change of the microstructure during the heating or cooling process. High-temperature laser confocal microscopy (CLSM) can solve these problems to some extent. In this experiment, in order to observe the changes of matrix structure and interface of WC/HSS composite during heating, the microstructure of the composite during heating was observed and analyzed by VL2000DX ultra-high temperature laser confocal microscope.

Figure 12 shows that the samples of tungsten carbide/high strength steel composites with particle size of 200 nm and sintering temperature of 1300 °C were observed in situ by CSLM during the heating process. Variable temperature was shown in the continuously recorded pictures. Figure 12a–i shows the microstructural changes of the composites from room temperature 25 °C to 1320 °C. When the temperature rises to 1002 °C, the structure at the interface between high strength steel matrix and tungsten carbide begins to melt, as shown in Figure 12b. When the temperature rises to 1002–1151 °C, the precipitation of black dot carbides in the high-strength steel matrix begins to increase, as shown in Figure 12b–e. When the temperature rises to 1202 °C, the second phase precipitates at the grain boundary of high strength steel matrix, and begins to melt from the carbide at the grain boundary, and the high-strength steel matrix begins to change from solid to a small amount of liquid, as shown in Figure 12f. As the temperature rises to 1202–1320 °C, the high-strength steel matrix gradually melts, and liquid content gradually increases, showing a molten state, as shown in Figure 12f–i. In addition, when the temperature rises to 1202 °C, the molten iron in the high-strength steel matrix begins to diffuse from the interface to the tungsten carbide structure, as shown in Figure 12f, and the diffusion layer widens gradually with the increase of temperature.

Figure 13 shows that the tungsten carbide/high-strength steel composite samples prepared at the sintering temperature of 1300 °C and a particle size of 500 nm were observed by CSLM in situ during the heating process, and the variable temperature was shown in the continuous recorded pictures. Figure 13a–i shows the changes of the microstructures of the composites during the process from room temperature 23 °C to 1346 °C. When the temperature rises to 1051−1101 °C, the surface of high strength steel begins to precipitate fine granular carbide, as shown in Figure 13b,c. When the temperature is 1150 °C, the second phase precipitates at the grain boundary of high-strength steel matrix, as shown in Figure 13d. When it rises to 1150−1300 °C, the granular carbide begins to increase and the grain boundary deepens, as shown in Figure 13d–g. When the temperature is between 1320 °C and 1346 °C, the high-strength steel structure begins to melt and the surface liquid gradually increases, as shown in Figure 13h,i. In addition, during the whole heating process, the phenomenon that the high-strength steel matrix diffused from the interface to the tungsten carbide structure was not found, which is consistent with the results of the previous mid-surface scanning analysis.

## 4. Conclusions

(1)Ultrafine tungsten carbide/high-strength steel composites for micro-components were successfully fabricated using the cold pressing-vacuum sintering method with special dies, and the combination of powder and solid was realized.(2)With the decrease of WC particle size, the average grain size of WC decreases. Meanwhile, WC density tended to increase, when the particle size of WC powder was 100 nm, the density reached 91.22%.(3)The hardness of the WC side of the composite increases with the decrease of WC particle size. When the particle size of WC powder is 100 nm and sintering temperature is 1320 °C, the hardness of WC reaches about 1680 HV0.1. The hardness of WC near the bonding interface is higher than that at other locations. The high-speed steel material in the core changes little with the WC particle size, which is about 500 HV0.1.(4)When the particle size of the WC powder particles is 200 nm, the tungsten carbide and the high-strength steel form a transition layer of a certain width at the interface, and the combination of the composite materials represents metallurgical bonding. The iron (Fe) in the high-strength steel matrix diffuses into the contacted tungsten carbide structure, which produces a certain width of the fusion layer, and the composite interface is relatively well bonded.(5)When the average particle size of the WC powder is 200 nm, in the transition region, the diffusion of the W, Fe and Co elements is significant at the interface. When the particle size of the tungsten carbide particles increases, the tendency towards element diffusion decreases. When the particle size of the WC powder particles is 500 nm, no significant diffusion of Fe and W elements occurs at the interface of the composite, and only a certain degree of diffusion of the Co element occurs.

## Figures and Tables

**Figure 1 materials-12-04164-f001:**
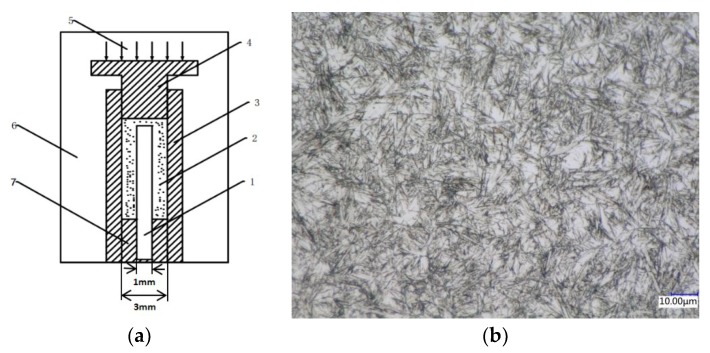
Schematic and physical drawings of cold pressing-vacuum sintering mold (**a**). Parts of the mold are: 1, high-strength steel core; 2, WC and Co mixed powder; 3, mold matrix; 4, punch; 5, constant pressure; 6, vacuum environment; 7, base. (**b**) The microstructure of M2 high speed steel.

**Figure 2 materials-12-04164-f002:**
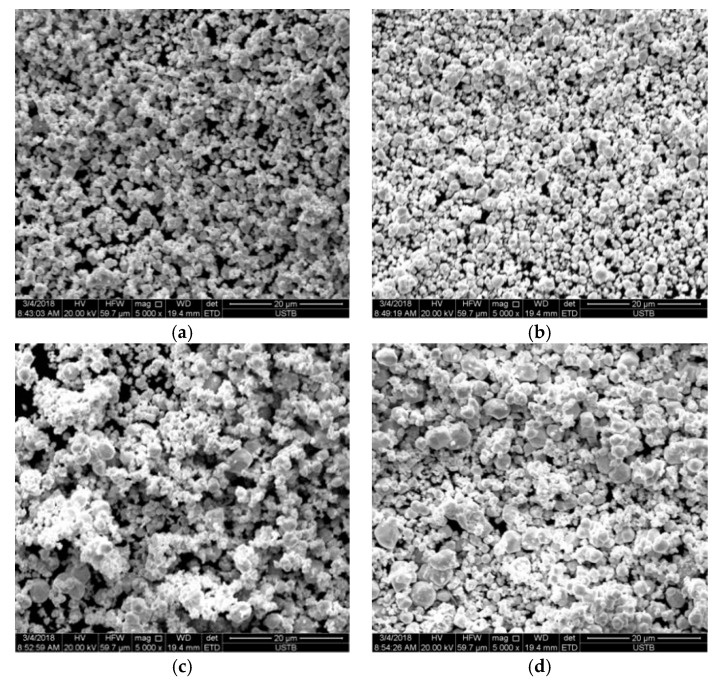
SEM image of original WC powder morphology with different particle sizes (**a**) 100 nm; (**b**) 200 nm; (**c**) 300 nm; (**d**) 500 nm.

**Figure 3 materials-12-04164-f003:**
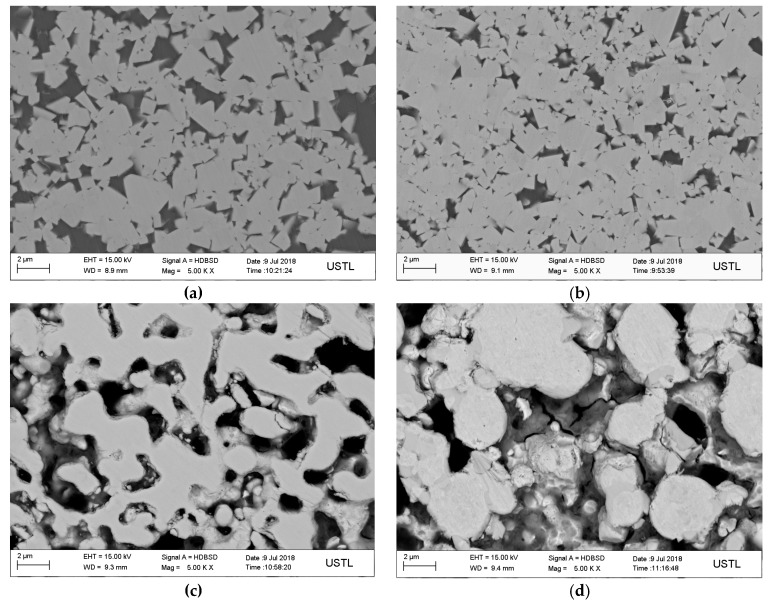
The WC-10Co SEM image under different particle sizes (**a**) 100 nm; (**b**) 200 nm; (**c**) 300 nm; (**d**) 500 nm.

**Figure 4 materials-12-04164-f004:**
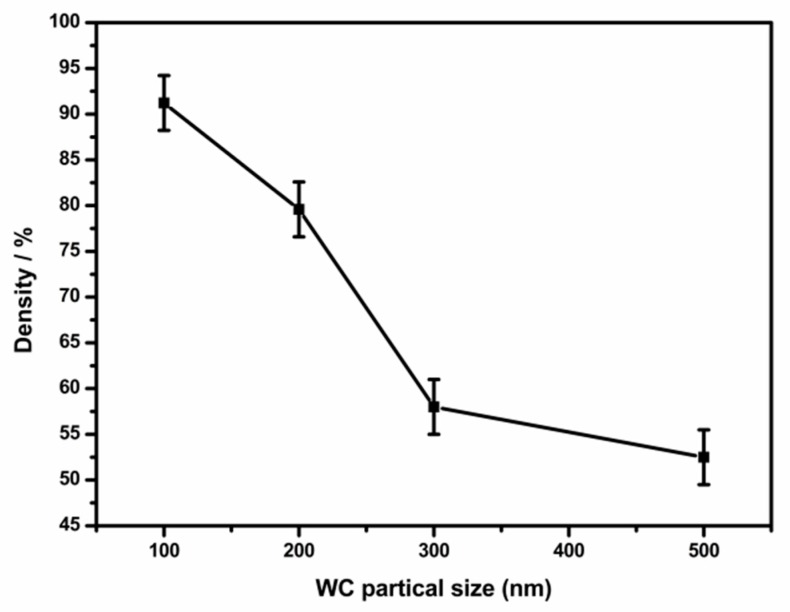
Density of WC-10Co under different particle sizes.

**Figure 5 materials-12-04164-f005:**
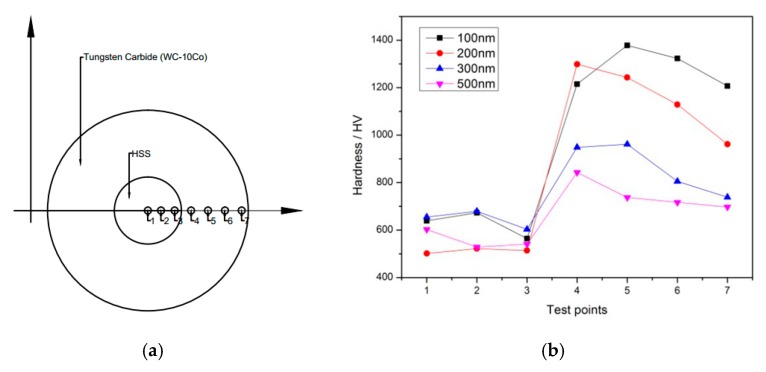
The measurement of microhardness distribution of composites: (**a**) The measurement position of microhardness; (**b**) Microhardness distribution of composites with sintering temperature of 1300 °C under different particle sizes.

**Figure 6 materials-12-04164-f006:**
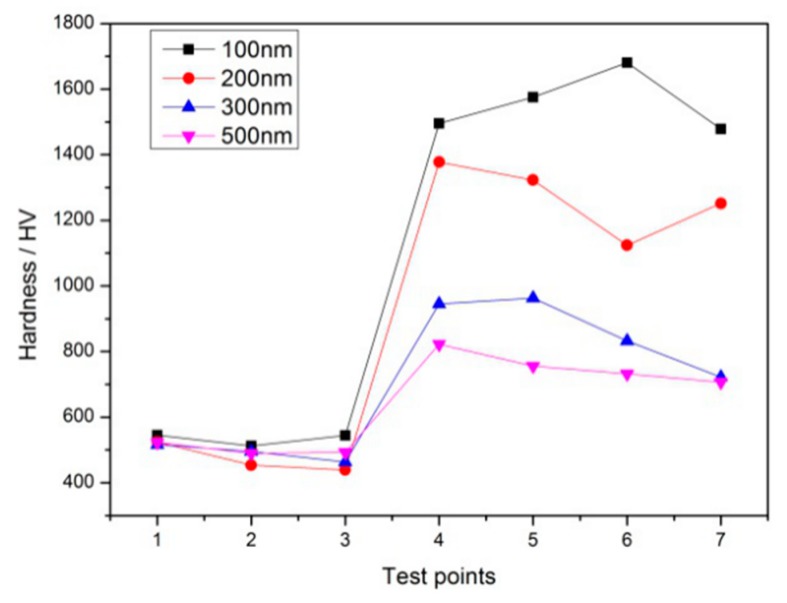
Microhardness distribution of composites with sintering temperature of 1320 °C under different particle sizes.

**Figure 7 materials-12-04164-f007:**
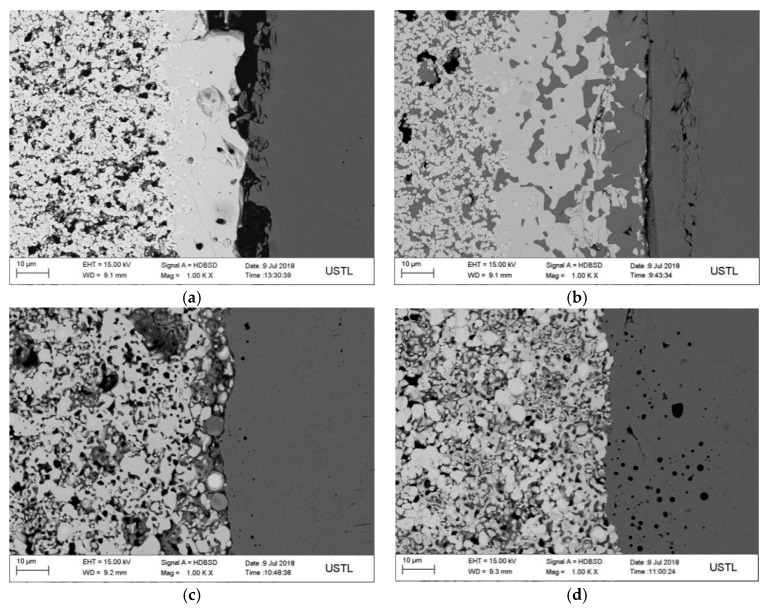
SEM photos of interface of WC/high-strength steel composites with different particle sizes: (**a**) 100 nm; (**b**) 200 nm; (**c**) 300 nm; (**d**) 500 nm.

**Figure 8 materials-12-04164-f008:**
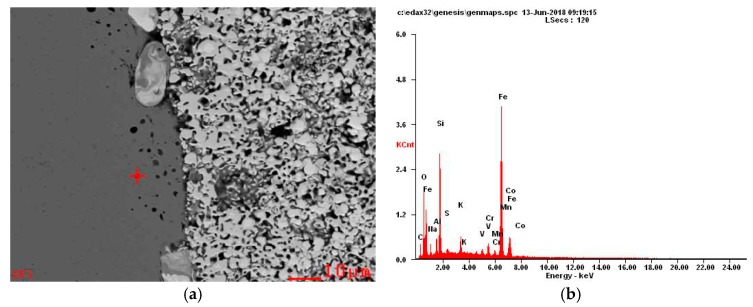
Interfacial energy spectrum analysis of WC/high-strength steel composites with particle size of 500 nm.

**Figure 9 materials-12-04164-f009:**
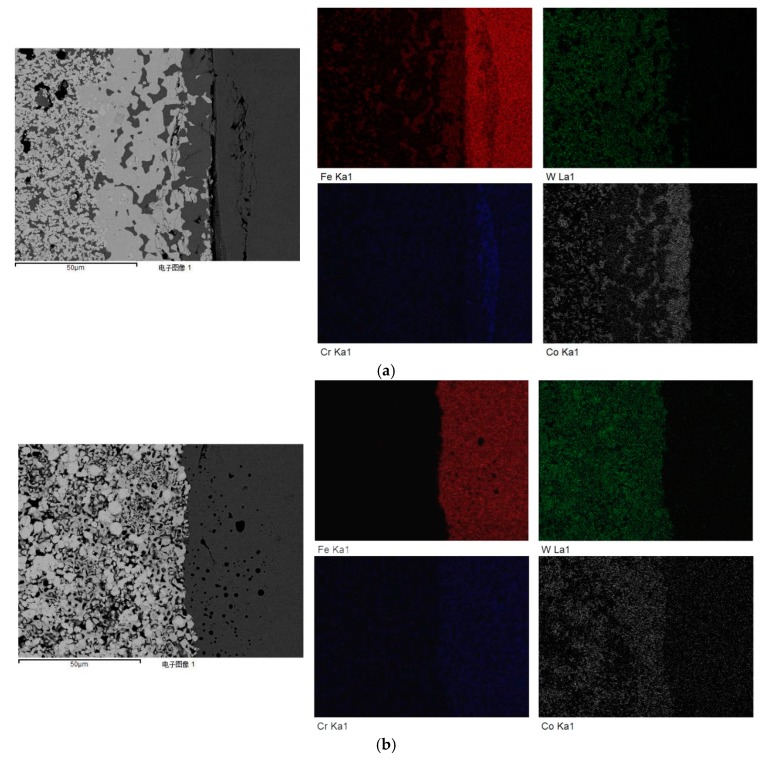
Surface distribution (EDS) of various elements at the composite interface of WC/high-strength steel under different particle sizes: (**a**) 200 nm; (**b**) 500 nm.

**Figure 10 materials-12-04164-f010:**
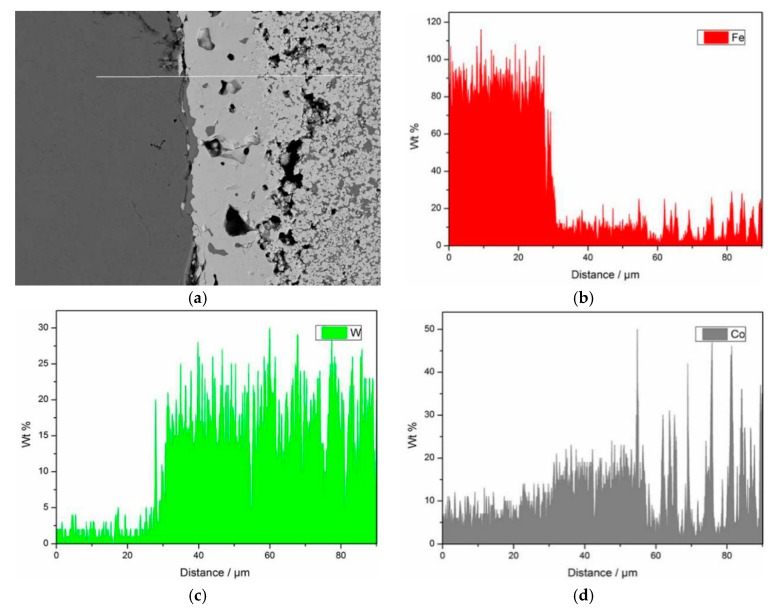
Line scanning of composite interface with WC particle size of 200 nm.

**Figure 11 materials-12-04164-f011:**
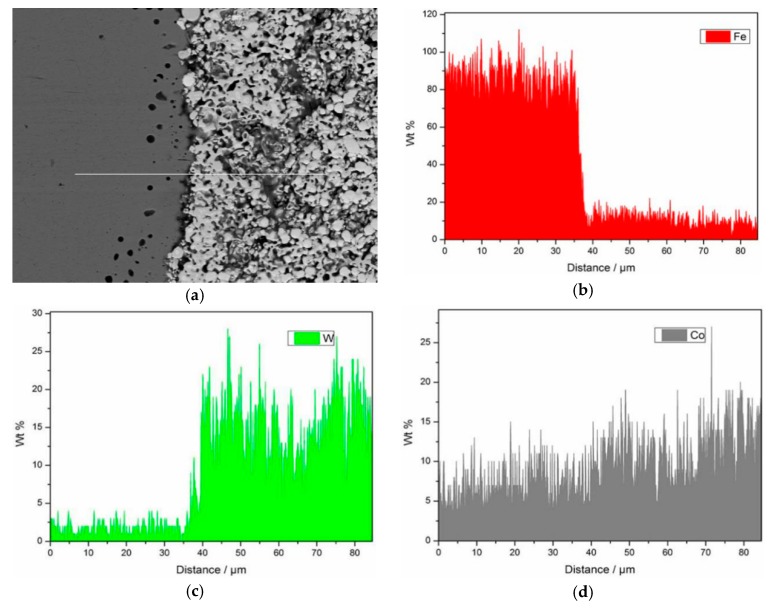
Line scanning of composite interface with WC particle size of 500 nm.

**Figure 12 materials-12-04164-f012:**
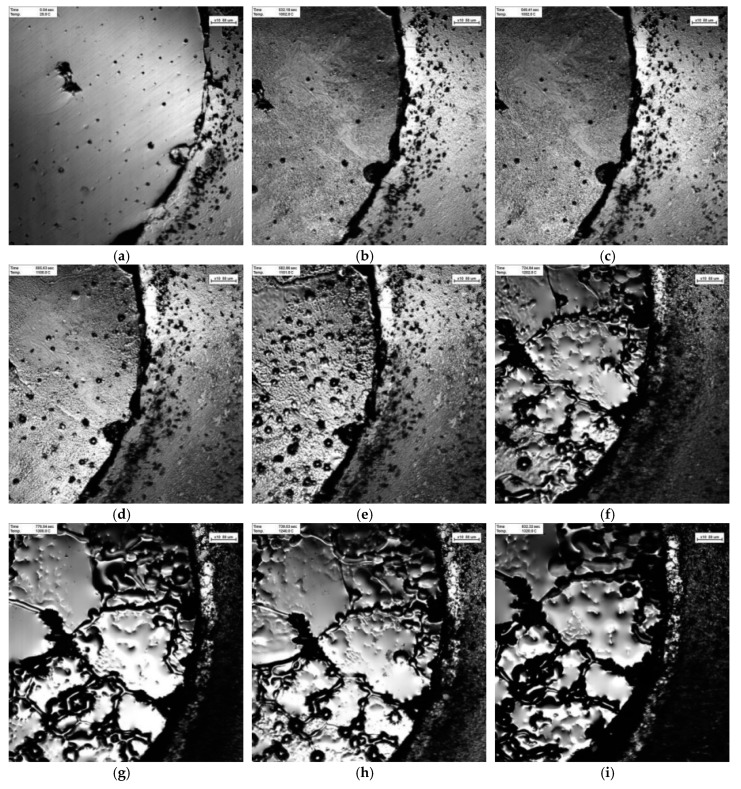
In-situ observations using high-temperature laser confocal microscopy (CSLM) during temperature rise of WC/high strength steel composite with particle size of 200 nm. (**a**) 25 °C; (**b**) 1002 °C; (**c**) 1052 °C; (**d**) 1100 °C; (**e**) 1151 °C; (**f**) 1202 °C; (**g**) 1252 °C; (**h**) 1300 °C; (**i**) 1320 °C.

**Figure 13 materials-12-04164-f013:**
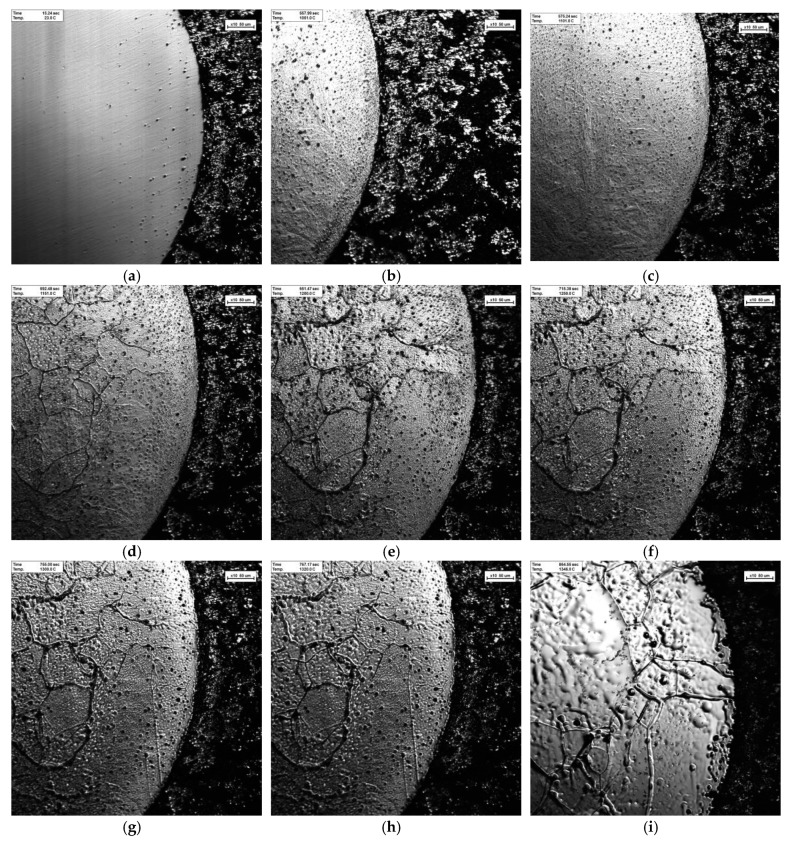
In-situ observation using CSLM during temperature rise of WC/high-strength steel composite with particle size of 500 nm. (**a**) 23 °C; (**b**) 1051 °C; (**c**) 1101 °C; (**d**) 1150 °C; (**e**) 1200 °C; (**f**) 1250 °C; (**g**) 1300 °C; (**h**) 1320 °C; (**i**) 1346 °C.

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
