# Peer review of "Effect of Particle Size on Microstructure and Element Diffusion at the Interface of Tungsten Carbide/High Strength Steel Composites"

_materials, 2019, doi:10.3390/ma12244164_

Round 1

Reviewer 1 Report

General Evaluation

The paper is an interesting study of WC/Co-HSS steel composite material characterization using mainly SEM/EDS analysis, microhardness testing and Laser Confocal Microscopy. The study is mainly focused on the influence of initial WC particle size on the bonding quality, elemental diffusion and sintered ceramic properties (density and hardness). The paper is comprehensively presented, well organized and within the scope of the Journal. Specific suggestions for further improvement are following.

Technical/Scientific Comments

Experimental Part

Density measurement technique (presented in Results, see Fig. 4) seems that it is not referred in Experimental part. Please provide a more detailed drawing (with dimensions) for Figure 1. Initial microstructure of M2 steel could be helpful if it is included (if it is available).

Results and Analysis

Truncation method is not well understood (L134). Please specify. Fig. 4: are there any error bars? L128: please explain and justify the maximum agglomeration for 300 nm particle size. How this was assessed? Microhardness evolution plots (Figs. 5-6) should be differently presented. X-axis should be presented as distance (e.g. from surface?) and the WC/Steel interface should be indicated, together with both material sides. Caption of Fig. 6 indicates that sintering temperature was 1300oC while the text referred to 1320oC (L186). Please check. Indicate the sintering temperature in the relevant Figures, e.g. see Figs. 7-9. The details in elemental maps in Figs. 8-9 are barely seen. “Melting” is frequently referred in the text (see for example L288). Please confirm whether melting occurs (e.g. incipient melting, above solidus temperature) or it is just high temperature diffusion. L303: 500 nm should be (instead of 200 nm), referring to Fig. 13.

Grammatical/Spelling Comments

The text need to be carefully corrected, concerning language and grammar in order to be suitable for publication. There are also several “mismatches” and discrepancies in symbols and statements which demand further revisions. For example:

There are spelling errors in affiliations (e.g. no. 4). Sections need to start from 1 and not from 0. Reference 13: please correct the names of the authors. Units are not correctly expressed (s lower case for time instead of S, oC for Temperature instead of C). Also, Cobalt chemical symbol is Co and not CO. In L96, MPa should be written instead of Mpa. As a general suggestion, keep one single space between the numerical expression and the units of physical measurements (apart from temperature). L146. Fig. 4 is a graph or plot and not a micrograph. L204. In Figure 7, micrographs are presented and not graphs. L208. Fe atoms (instead of elements). In L291, L308; “the grain boundary … begins to precipitates” in not correct expression. Please rephrase and describe the phenomenon more clearly, e.g. carbides tend to precipitate at grain boundaries.

Reviewer 2 Report

Research topic deserves attention because of its relevance. The development of composite materials technology requires solving a whole range of material science problems. Among these problems is the analysis of the interaction of components in complex metal matrix compositions such as tungsten carbide / high strength steel.

A large number of parameters affect the nature of the interaction of hard inclusions and the steel core during sintering.

The authors proposed to consider the effect of particle sizes of tungsten carbide on the formation of the transition zone "tungsten carbide/steel core" and to study its structural characteristics.

As a remark, it should be noted that the authors have indicated dimensions tungsten carbide fractions, but do not specify the size of the cobalt powder used.

It was indicated that the ratio of tungsten carbide to cobalt is 9:1, but there is no clarification in which units — volume or mass.

Authors also necessary to use when translating the more appropriate English terms (e.g., use "assemblage" instead of "reunion" (line 127) or "mold matrix" is preferable than "female die", line 110, and "dependence curve" instead of "micrograph", line 146).

It is necessary to pay attention to the grammar of English sentences (lines 74-76, lines 126-127, 162-163, 175-177, and 206-209).

In addition, there are a number of typos (lines 92, 123-126, 138, 203, 233-234, 244-245).

Manuscript text adjustment required.

Round 2

Reviewer 1 Report

The authors addressed satisfactorily the review comments and, therefore, the paper is recommended for publication.